# Comparative Proteome Profiling of Extracellular Vesicles from Three Growth Phases of *Haematococcus pluvialis* under High Light and Sodium Acetate Stresses

**DOI:** 10.3390/ijms25105421

**Published:** 2024-05-16

**Authors:** Qunju Hu, Yuanyuan Wang, Chaogang Wang, Xiaojun Yan

**Affiliations:** 1College of Marine Science and Technology, Zhejiang Ocean University, Zhoushan 316022, China; huqunju2022@zjou.edu.cn (Q.H.); y704379952@163.com (Y.W.); 2Shenzhen Engineering Laboratory for Marine Algal Biological Development and Application, College of Life Sciences and Oceanography, Shenzhen University, Shenzhen 518060, China

**Keywords:** extracellular vesicles, *Haematococcus pluvialis*, different growth phases, proteome profile, biological functions

## Abstract

Extracellular vesicles (EVs) are nano-sized particles involved in intercellular communications that intrinsically possess many attributes as a modern drug delivery platform. *Haematococcus pluvialis*-derived EVs (HpEVs) can be potentially exploited as a high-value-added bioproduct during astaxanthin production. The encapsulation of HpEV cargo is a crucial key for the determination of their biological functions and therapeutic potentials. However, little is known about the composition of HpEVs, limiting insights into their biological properties and application characteristics. This study examined the protein composition of HpEVs from three growth phases of *H. pluvialis* grown under high light (350 µmol·m^−2^·s^−1^) and sodium acetate (45 mM) stresses. A total of 2038 proteins were identified, the majority of which were associated with biological processes including signal transduction, cell proliferation, cell metabolism, and the cell response to stress. Comparative analysis indicated that *H. pluvialis* cells sort variant proteins into HpEVs at different physiological states. It was revealed that HpEVs from the early growth stage of *H. pluvialis* contain more proteins associated with cellular functions involved in primary metabolite, cell division, and cellular energy metabolism, while HpEVs from the late growth stage of *H. pluvialis* were enriched in proteins involved in cell wall synthesis and secondary metabolism. This is the first study to report and compare the protein composition of HpEVs from different growth stages of *H. pluvialis*, providing important information on the development and production of functional microalgal-derived EVs.

## 1. Introduction

Extracellular vesicles (EVs) are membrane bilayer-bound spherical vesicles encapsulating or containing parent-cell-specific biological molecules, such as lipids, proteins, nucleic acids, and metabolites, which are potentially produced and secreted by cells into the extracellular space [1,2]. EVs are engaged in intercellular communication by transporting various bioactive molecular cargoes and delivering them to local or distant recipient cells [3]. EVs are also involved in maintaining cellular homeostasis by altering their metabolism to compensate for cells’ responses to stress [2,3] and play a novel physiological role in the maintenance of cellular integrity [4] and organismal homeostasis [4]. In addition, EVs can influence ecosystem function and determine the structure and composition of microbial populations [5,6,7]. Moreover, EVs are inherently packed with therapeutic molecules from their source cell and can also deliberately sort molecules of interest within the lumen or on the surface through exogenous or endogenous means [8,9], along with further editable and targetable capabilities [9,10,11,12]. These features endow those naturally secreted nanoparticles with the potential to act as innate drug delivery vehicles and next-generation therapeutics [3,11,12,13]. Natural EVs have several advantages over conventional synthetic nanocarriers [11,12], including biocompatibility [11], suitability for modifications [12,13], and capability to improve in vivo circulation, as well as the potential for increased stability and efficacy of drugs from enhanced cellular uptake because of the cell-like membrane topology along with the lumen of EVs [14,15]. Harnessing EVs for therapeutic applications primarily relies on their biomolecular composition, which is not random, and each type of EV cargo delivers specific molecular messages [3,8,12,16]. Proteins are the ultimate executors of gene function [17]. Therefore, the protein cargoes of EVs confer them particular physiological properties, e.g., molecular binding and surface fusion [16]. From a drug delivery perspective, the complexity of EV protein cargo needs to be elucidated and addressed in all characterization and production processes via comprehensive (multi)omics studies [11]. High-sensitivity mass spectrometry-based proteomic datasets and protein interaction networks have established significant relationships among EV proteins and revealed significant changes in protein accumulation under various physiological and pathological conditions, thereby improving the comprehension of EV biogenesis and pathophysiological roles [18].

Until now, the biological functional roles of EVs have been studied in different organisms [19]. In recent years, academic and industrial interests in microalgae as a novel natural source of EVs have increased [20,21,22,23,24,25,26,27]. It was suggested that the secretion of microalgae EVs is an evolutionary conserved trait, and some studies demonstrated that microalgae are promising producers of EVs that could be used as sustainable and natural nanocarriers of bioactive compounds [21,22]. For example, Picciotto et al. [22] found six best-scoring microalgal strains for the production of EVs, including the freshwater glaucophyte *Cyanophora paradoxa*, marine chlorophyte *Tetraselmis chuii*, marine dinoflagellate *Amphidinium* sp., and rhodophyte *Rhodella violacea*, among eighteen studied microalgal stains. Differential ultracentrifugation and tangential flow fractionation were identified as promising methods for the large-scale production of EVs from microalgae [21,22]. Compared with the use of other sources for EV production, the use of microalgae as a natural source for EVs offers several advantages, including sustainability, scalability, controllability, and renewability [20,21,22]. Microalgae-derived EVs can be used as tailor-made high-value-added bioproducts in different industrial sectors, such as nanomedicine, nutraceuticals, and cosmetics [21,23,24]. In addition, the natural and sustainable origin of microalgae-derived EVs grants them greater societal acceptance as a source for formulation preparations [21,24]. Previous reports have demonstrated that microalgal-derived EVs are non-cytotoxic and can be taken up by different cellular systems of human cells, the model organism *Caenorhabditis elegans* (Nematoda), and mice. With the advantages of unparalleled biocompatibility and unique tropism and bioactivities, they are suitable as innate antioxidant and anti-inflammatory effectors [21,25,26]. They also have the capacity to be produced at a mass scale with a renewable and sustainable bioprocess, suggesting that microalgae can act as a novel biofactory for biocompatible and bioactive extracellular vesicles [22,26]. In an ideal situation, this technology offers microalgae-derived EVs potential applications in biological pharmaceutical, cosmetics, and nutraceuticals market sectors, in which they eventually act as delivery vehicles for therapeutic agents.

The unicellular freshwater microalga *Haematococcus pluvialis* has been considered the best natural source for astaxanthin production for a long time [28]. Moreover, *H. pluvialis* also simultaneously accumulates diverse metabolites with high commercial value during astaxanthin accumulation in the red stage, including lipids, carbohydrates, and proteins, making it an attractive feedstock for multiple-product biorefining toward a higher commercial realization [28,29,30]. Therefore, *H. pluvialis*-derived EVs (HpEVs) can also be obtained through integration as a novel production process because HpEVs can be harvested from the culture media of *H. pluvialis*, which will not affect the biomass. The potential exploitation of HpEVs as tailor-made high-value-added bioproducts of *H. pluvialis* biomass will be a noteworthy attempt for the future development of *H. pluvialis* biotechnology. For this, fundamental knowledge of HpEV physiology, including their morphology and function in microalgal growth and metabolism, must be initially obtained. Understanding the composition of HpEVs is the key to understanding their contributions to cellular and molecular regulation in *H. pluvialis*. However, to our best knowledge, the proteomics basis of HpEVs remains unknown. Proteomic analysis has been widely used to characterize various types of protein cargoes in EVs [1,11,31]. In this study, proteomic analyses were performed to characterize the protein cargoes of EVs derived from *H. pluvialis* at different growth phases, including the green vegetative motile cell stage, green nonmotile cell stage, and red nonmotile cyst stage, using a high-resolution liquid chromatography–tandem mass spectrometry (LC-MS/MS)-based protein quantification approach with isobaric multiplex tandem mass tags (TMTs). Classification of the subcellular localization of the quantified proteins, protein-related biological processes, and metabolic pathways is necessary for the identification of specific features of HpEVs based on bioinformatics analysis. Furthermore, comparative research on these proteomes will unveil the distinctive functions of differential HpEVs along with the growth and astaxanthin accumulation in *H. pluvialis*. This study provides novel insights into the protein composition of HpEVs via proteomic strategies and depicts the potential mechanism of cell-to-cell communication. The results provide a basis for further research on the potential application of HpEVs as high-value coproducts of *H. pluvialis* biomass.

## 2. Results

### 2.1. Collection of HpEVs from the Culture Medium of H. pluvialis Suffering from High Light and High Sodium Acetate Stresses

As shown in Figure 1, the color of the HpEV pellets isolated from the culture medium of *H. pluvialis* at three growth phases was consistent with that of microalgal cells observed under the microscope. All these HpEVs were cup-shaped membrane bilayer-bound spherical or quasi-spherical nanovesicles with negative staining, revealing that the three HpEVs had similar morphology and size (Figure 1c). In this study, the protein concentrations of the isolated HpEVs were 37.85 ± 2.03 μg·mL^−1^, 44.57 ± 4.44 μg·mL^−1^, and 188.66 ± 22.87 μg·mL^−1^ in HpEVs-1, HpEVs-2, and HpEVs-3, respectively (Table 1). The results were consistent with our previous report that HpEVs were produced and accumulated in culture media [27]. The HpEVs were then used for the proteomics analysis.

### 2.2. Protein Identification and Quantification

In this study, we measured proteins in the HpEV pellets and also compared their abundance in HpEVs-1, HpEVs-2, and HpEVs-3. The SDS-PAGE analysis of proteins detected with silver staining revealed clear, homogenous, and undegraded protein in each HPEV sample (Figure 2a). The protein bands of the two replicates of each HPEV sample were similar, and no significant differences were observed among different HpEVs. This result was consistent with that obtained using a TEM, wherein the morphology of the HpEVs from three phases was remarkably consistent with each other. The proteins of the purified HpEVs were digested, and then the protein samples were identified using TMT labeling. A total of 4113 distinct proteins were identified across all HPEV samples. Distinct proteins were defined as those with one or more unique peptides. The full list of the identified proteins is provided in Appendix A. The results further indicated that HpEVs contain substantial amounts of proteins, and the HpEVs extracted from three phases exhibited similar physiological and biochemical features. Most of the identified peptides were 7–16 amino acids in length (Figure 2b). Furthermore, 41.4% of identified master proteins had a protein sequence coverage rate higher than 10% (Figure 2c). The quantifiable proteins of HpEVs were defined as those with two or more unique peptides and, finally, 2038 quantifiable proteins were identified in all HpEVs (Appendix A). The mass of the most identified proteins ranged between 10 and 60 kDa, and only approximately 0.75% of identified proteins had a molecular weight greater than 200 kDa. The results demonstrated the optimal quality and high credibility of LC-MS/MS data obtained in this study.

### 2.3. Proteome Profiles, Molecular Functions, and Biological Processes of Proteins in HpEVs

To answer whether the HpEVs derived from three different growth phases of *H. pluvialis* have similar characteristics, their proteome profiles were examined. Based on the comparison of the PSM value, the HpEVs derived from different growth phases of *H. pluvialis* shared similar proteins, but the abundance of these proteins varied. The identified proteins were further investigated to determine their subcellular location (Figure 3a), and the results revealed that these proteins were localized in the chloroplast (45.34%), cytoplasm (23.45%), nucleus (10.79%), mitochondria (7.02%), plasma membrane (6.53%), extracellular space (2.26%), vacuolar membrane (1.62%), endoplasmic reticulum (1.12%), and other areas (1.03%). These results demonstrated that HpEV proteins are likely derived from plastids and multivesicular bodies.

To examine the molecular function of the HpEV proteins, GO annotation and enrichment to the biological processes (BPs), cell components (CCs), and molecular functions (MFs) were performed (Figure 3b). According to GO terms related to biological processes (BPs), the HpEV proteome was highly involved in cellular metabolic processes (11.58%), organic substance processes (11.51%), primary metabolic processes (10.61%), nitrogen compound metabolic processes (9.32%), and biotic and abiotic stress responses (18.96%). The HpEV proteome was highly enriched in proteins involved in biotic and abiotic stress responses, with approximately 18.96% of the HpEV proteins being involved in such processes. The major cellular component (CC)-related GO terms were intracellular (19.79%), intracellular organelle (17.52%), membrane-bounded organelle (16.87%), cell periphery (4.61%), endomembrane system (4.52%), and non-membrane-bounded organelle (4.52%), indicating that HpEV proteins may be derived from corresponding components of the microalgal cells. Regarding the main molecular function of HpEV proteins, 53.87% of them were associated with MF binding, mainly including organic cyclic compound binding (12.70%), heterocyclic compound binding (12.50%), and protein binding (10.46%). Several HpEV proteins were associated with enzymes (29.48%), including hydrolase (10.39%), transferase (8.70%), and oxidoreductase (6.11%). The results indicated that HpEVs might be recognized by cells and bound to specific sites on the cell membrane, resulting in their secretion or ingestion by donor or receptor cells. The absorbed HpEVs may play roles in the regulation of metabolic processes and enzyme functions, thereby contributing to the growth, defense, and stress adaptation of the receptor cells.

Next, Clusters of Orthologous Groups (COG) classification revealed that the putative proteins were functionally classified into at least 23 molecular families, including “post-translational modification, protein turnover, chaperones”, “translation, ribosomal structure, and biogenesis”, “intracellular trafficking, secretion, and vesicular transport”, and “signal transduction mechanisms” (Figure 3c). These results revealed that HpEVs might play roles in information storage and processing, cellular processes and signaling, and cell motility and metabolism of the microalgal cells.

The KEGG enrichment analysis demonstrated that the most enriched pathways were related to cell growth, ribosome, proteasome, oxidative phosphorylation, aminoacyl-RNA biosynthesis, phagosome, and carbon fixation in photosynthetic organisms (Figure 3d). These pathways reflect growth characteristics, such as the regulation of cell division, cell wall construction, and protein biosynthesis in general. A few proteins involved in SNARE interactions in the vesicular transport pathway were detected, which indicated that these proteins might play roles in mediating HPEV transportation. The results indicated that these pathways respond to substantial alterations in the cellular metabolic programs, leading to the transportation of HpEVs in and out of microalgal cells.

### 2.4. The Growth Stage Significantly Affects the Proteome of HpEVs

To gain insight into the molecular changes associated with the growth stage, we further examined the protein composition of HpEVs from three growth phases of *H. pluvialis*. As shown in the hierarchical cluster heatmap (Figure 4a), all 2038 identified proteins were commonly found in all samples but with separations. HpEVs-1 and HpEVs-2 were aggregated into a cluster, while HpEVs-3 was aggregated into another cluster, revealing the growth stage-specific protein alteration. It was indicated that the HpEVs within the clusters had a high similarity, and those belonging to different clusters had remarkable differences. To further determine the effect of the growth stage on the protein composition in HpEVs, a principal component analysis (PCA) of the 2038 identified proteins was performed. The results revealed that the first two principal components explained 79.4% of data variability and each accounted for 58.4% and 21.0% of the total variance (Figure 4b). These results suggested that the HpEV samples can be divided into distinguished clusters based on their original cells, referring to different growth phases of *H. pluvialis* in this study. Overall, the results of the statistical analysis of the heat map and PCA were highly consistent, indicating that the objects to be analyzed could be divided into two clusters.

### 2.5. DAPs and Bioinformatics Analysis

In this study, proteins with a fold-change > 1.5 and *p*-value < 0.05 (*t*-test) between different HpEVs were defined as DAPs, which were compared and analyzed in different HpEVs (including HpEVs-1, HpEVs-2, and HpEVs-3) (Figure 4c). Of these proteins, a total of 63, 217, and 260 proteins were significantly upregulated, and a total of 99, 184, and 290 proteins were significantly downregulated in the comparisons of HpEVs-2 versus HpEVs-1, HpEVs-3 versus HpEVs-2, and HpEVs-3 versus HpEVs-1, respectively. A Venn diagram of the DAPs between different phases was constructed, and the results revealed the shared and unique DAPs among the comparison groups (Figure 4d).

#### 2.5.1. GO Annotation of DAPs

To evaluate the functional categories of HpEVs isolated from different growth phases of *H. pluvialis*, three groups of DAPs were annotated and enriched in the GO database to the three main categories including BP, CC, and MF. It was found that a high percentage of DAPs of the three groups were categorized in the BP categories of “cellular metabolic process”, “organic substance metabolic process”, “primary metabolic process”, “nitrogen compound metabolic process”, and “response to stress or stimulus” (Figure 5a). In the comparison of HpEVs-2 versus HpEVs-1, a much higher percentage of DAPs were annotated to GO categories, such as “response to stress”, “response to abiotic stimulus”, “cellular component biogenesis”, “regulation of biological quality”, “response to endogenous stimulus”, and “cell communication”, compared with that of the comparison HpEVs-3 versus HpEVs-2. Moreover, a lower relative percentage of DAPs were annotated to the categories “organic substance metabolic process”, “primary metabolic process”, “regulation of the biological process”, and “biosynthesis process”. The evaluation of CC categories (Figure 5b) revealed that a high percentage of DAPs were categorized as “intracellular”, “intracellular part”, “intracellular organelle”, “membrane-bounded organelle”, and “intracellular organelle part”. Compared with the HpEVs-3 versus HpEVs-2 group, a much higher relative percentage of DAPs in the HpEVs-2 versus HpEVs-1 group were annotated to the categories “envelope”, “membrane protein complex”, and “photosynthetic membrane”. A lower percentage of DAPs in the HpEVs-2 versus HpEVs-1 group were annotated into categories such as “membrane-bounded organelle”, “organelle lumen”, and “endomembrane system”. The analysis of GO-MF categories (Figure 5c) revealed that a high percentage of DAPs were categorized as “organic cyclic compound binding”, “heterocyclic compound binding”, “ion binding”, “hydrolase activity”, and “protein activity”. And a much higher percentage of DAPs related to “ion binding”, “cofactor binding”, and “pigment binding” were categorized in the HpEVs-2 versus HpEVs-1 group compared with the HpEVs-3 versus HpEVs-2 group. A lower percentage of DAPs related to “protein binding”, “transferase binding”, “oxidoreductase activity”, and “transmembrane transporter activity” was categorized in the HpEVs-2 versus HpEVs-1 group.

#### 2.5.2. COG Annotation of DAPs

Based on COG annotation, DAPs were classified into four categories, including metabolism, cellular process and signaling, information storage and processing, and unknown function (Figure 5d). In addition, most DAPs were assigned to the functional categories “energy production and conservation”, “post-translational modification, protein turnover, chaperones”, and “translation, ribosomal structure, and biogenesis”. Further investigation of the subcellular location of DAPs indicated that approximately 50% and 20% of the DAPs were localized in the chloroplast and cytoplasm, respectively. These results were consistent with those of COG functional annotation, revealing that most DAPs were classified in the energy production and protein processes categories. In addition, in the HpEVs-2 versus HpEVs-1 group, more DAPs were localized in the chloroplast than in the cytoplasm and plasma membrane, which were consistent with those of COG functional annotation, and more DAPs classified in the categories “energy production and conservation” and “translation, ribosomal structure, and biogenesis” in the HpEVs-2 versus HpEVs-1 group were observed, while more in the category “post-translational modification, protein turnover, chaperones” in the HpEVs-3 versus HpEVs-2 group. In summary, more proteins involved in energy production and translation functions were observed in the HpEVs extracted from the cells changing from the green vegetative motile cell stage to the green nonmotile cell stage (HpEVs-1 to HpEVs-2), and more proteins involved in protein post-translational modification were observed in the HpEVs extracted from the cells changing from the green nonmotile stage to the red nonmotile cyst stage (HpEVs-2 to HpEVs-3). These results indicate that under stress conditions, *H. pluvialis* cells sort different proteins into HpEVs, resulting in the fact that HpEVs secreted by cells at the early growth phase involved in energy production and translation and the HpEVs secreted by cells at the late growth stage were mainly involved in protein post-translational modification.

To further investigate if the DAPs were involved in substantial changes in metabolic activities during growth and astaxanthin accumulation in *H. pluvialis*, the levels of DAPs in specific COG functional categories during the cell stage changes were studied. A total of eight specific secondary functional categories were analyzed, including “cell cycle control, cell division, chromosome partitioning”, “replication, recombination and repair”, “cell wall/membrane/envelope biogenesis”, “signal transduction mechanisms”, “lipid transport and metabolism”, “intracellular trafficking, secretion, and vesicular transport”, “secondary metabolite biosynthesis, transport, and catabolism”, and “astaxanthin biosynthesis” (Figure 6). During the screening of the functional categories, the mean value of the DAPs in the chosen categories was used to construct data curves for further comparison and analysis. The results revealed that more DAPs identified in the categories “intracellular trafficking, secretion, and vesicular transport”, “secondary metabolite biosynthesis, transport, and catabolism”, and the “astaxanthin biosynthesis” pathways were upregulated, whereas more DAPs identified in the categories “cell cycle control, cell division, chromosome partitioning” and “signal transduction mechanisms” were downregulated. In addition, four DAPs identified in the category “cell wall/membrane/envelope biogenesis” were all upregulated during the change from the green vegetative motile cell stage to the red nonmotile cyst stage (HpEVs-1 to HpEVs-3).

#### 2.5.3. Distribution of Enriched KEGG Pathways

KEGG enrichment analysis revealed DAPs were related to pathways involved in photosynthesis and secondary metabolism in *H. pluvialis* (Figure 5e and Appendix A). On the bases of the significance analysis (*p* < 0.05), a total of five specific KEGG pathways were detected, including “ribosome”, “photosynthesis—antenna proteins”, “oxidative phosphorylation”, “photosynthesis”, and “monoterpenoid biosynthesis” in the HpEVs-2 versus HpEVs-1 group (Appendix A). In the HpEVs-3 versus HpEVs-2 group, a total of three specific KEGG pathways were detected, namely, “citrate cycle (TCA cycle)”, “plant–pathogen interaction”, and “protein processing in the endoplasmic reticulum”. Of these, six and five pathways were upregulated and downregulated, respectively. In the HpEVs-3 versus HpEVs-1 group, a total of seven specific KEGG pathways were strongly enriched, namely, “biosynthesis of unsaturated fatty acids”, “ribosome”, “TCA cycle”, “fatty acid elongation”, “protein processing in the endoplasmic reticulum”, “plant–pathogen interaction”, and “spliceosome”. Of these, nine and seven pathways were upregulated and downregulated, respectively. The number of DAPs and upregulated and downregulated KEGG pathways are listed in Appendix A.

## 3. Discussion

It is generally believed that extracellular vesicles (EVs) are important mediators of intercellular communication via transferring a wide variety of molecular cargoes [1,8,13,32,33]. The property of EVs that carry parent-cell-specific signatures permits target-cell interactions in distinctly different manners [34]. Understanding this complexity and addressing its characterization and production processes is essential from the drug delivery perspective [11]. The protein components in EVs not only reflect the specific parent cell origins but also vary depending on the nature and biogenesis of different EV subpopulations [35]. Thus, attempts to analyze the protein cargo of EVs are essential for the understanding of their cellular functions and biological properties [30] and are also important in further research studies of their utilization [15]. *H. pluvialis*-derived EVs (HpEVs) could be considered tailor-made high-value-added bioproducts of astaxanthin or biomass, and their application may be a potential field of microalgae biotechnology [21,23,24].

In this study, HpEVs were isolated from three growth phases of *H. pluvialis*, including green vegetative cells, green nonmotile cells, and red cysts. The color of the HpEVs varied and was consistent with the physiological status of *H. pluvialis* cells, which might indicate that the microalgal cells sort various cargoes into HPEVs (Figure 1). A quantitative proteome analysis of these HpEVs was further performed, and the result suggested that there a large number of proteins were detected (Appendix A). Meanwhile, similarities and differences in protein species and their abundance or expression levels were determined in different HpEVs, providing insights into the composition and function of HpEVs (Figure 7, generated with BioRender, https://www.biorender.com/, accessed on 15 March 2024). Specifically, a total of 2038 ubiquitous proteins were found, which likely comprise the core proteome of HpEVs (Appendix A). The proteins were mainly localized in the chloroplast, indicating that these HpEVs were likely derived from plastids and multivesicular bodies with distinct functions (Figure 3) [5,32,36,37,38,39]. Most of them were typically associated with cell components and functions, such as structure, biogenesis, and trafficking (Figure 3) [7,32,39,40,41]. The biological process annotations revealed that HpEV proteins played a role in cellular metabolic processes, organic substance processes, primary metabolic processes, and biotic and abiotic stress response processes (Figure 3). Moreover, the molecular function annotations demonstrated that binding proteins also played a role in the vesicle-mediated transport of HpEVs by promoting the sorting of a subset of cargoes into HpEVs and the binding of HpEVs to cells (Figure 3) [3,39,41,42,43]. These findings suggested that the cellular communication of HpEVs might reply to the protein complements (especially the binding activity) and the receptors on the recipient cells [44]. These binding proteins might play important roles in the interaction between EVs and their recipient cells, through mechanisms that allow EV derivatives to bind to respective receptors on target cells as a ligand [11,45]. HpEV enzymes with various catalytic functions further regulate the growth, defense, and stress adaptation of receptor cells. The results of the COG annotation revealed that most HpEV proteins were annotated as signal transduction mechanisms, indicating that HpEVs may play important roles in information exchange among microalgal cells [5,21,46]. The KEGG pathway enrichment analysis revealed that HpEV proteins were involved in the growth and cellular metabolic processes of *H. pluvialis*, which also mediated HpEV transportation. HpEVs were also involved in the biological function of stress response mediation in the microalgal cells, a result consistent with that observed in bacterial EVs (Figure 3) [1,5,7,47,48]. Our findings shed light on the protein components in HpEVs and demonstrated their homogeneity among different HpEVs, which reflected the cell origin of these EVs and also implicated their cellular functions mainly involving signal transduction, cell proliferation, cell metabolism, and cellular stress responses. Furthermore, the homogeneity in the HpEV protein components might also have a profound impact on maintaining of the stability of HpEVs under stress conditions by preventing EV degradation [15,49].

The comparative proteomic analysis also revealed heterogeneous protein assembly among HpEVs from different growth phases of *H. pluvialis*, and DAPs were subsequently identified (Appendix A, Figure 4, Figure 5 and Figure 6). Interestingly, DAPs associated with biological processes were observed since DAPs responding to stress or stimulus accumulated much more during the stage change from green vegetative motile cells (HpEVs-1) to green nonmotile cells (HpEVs-2) than from green nonmotile cells (HpEVs-2) to red nonmotile cells (HpEVs-3). The result also indicated that the biotic and abiotic stress responses of HpEVs changed significantly in the early stage (Figure 4). The GO analysis of the higher percentage of DAPs observed in HpEVs-2 versus HpEVs-3 revealed that the functions of HpEVs in the late stage of the stress treatment were involved in metabolic processes (Figure 3). Further COG annotation revealed that most DAPs were involved in energy and protein production. More DAPs related to energy production and translation in the early stage of the cell response to stress conditions were observed in HpEVs-1 versus HpEVs-2. Moreover, more DAPs related to protein post-translational modification in the late stage of the cell response to stress conditions were observed in HpEVs-2 versus HpEVs-3 together with the microalgal cell adaption to stress conditions. Meanwhile, the majority of the DAPs were predicted to be localized in the chloroplast and cytoplasm, and a higher percentage of DAPs located in the chloroplast were observed in the HpEVs-2 versus HpEVs-1 group. More DAPs located in the cytoplasm and plasma membrane were observed in HpEVs-3 versus HpEVs-2 (Figure 5). The results of the COG annotation and subcellular location investigation were in accordance with chloroplast and cytoplasm functions. The analysis of the specific COG functional categories demonstrated that the functions of HpEVs involved in cell division and signal transduction were inhibited during the stress treatment, whereas the functions of HpEVs involved in cell wall biosynthesis, intracellular trafficking, and secondary metabolites were promoted during the same time. In the present study, the results of the KEGG enrichment analysis of the DAPs were similar to those of the GO and COG annotations (Figure 6). These results suggested that HpEVs from the early growth stage of *H. pluvialis* contained proteins associated with cellular functions of primary metabolites, cell division, and cellular energy metabolism, while HpEVs from the late growth stage of *H. pluvialis* were rich in proteins associated with cell wall synthesis and secondary metabolism (Figure 7). The above results were consistent with previous reports that EVs play essential roles in modulating metabolic homeostasis related to phase shifts [50,51]. In addition, the composition of EVs reflects the status of their parent cell, participates in metabolic homeostasis, and endows them with heterogeneous cellular functions [9,50,51]. The comparative analysis of the protein composition of HpEVs in different phases not only unveils the distinctive nature, biogenesis, and function of EVs over the life cycle of *H. pluvialis* but also provides insight into the potential application of HpEVs as a drug delivery platform.

## 4. Materials and Methods

### 4.1. Acquisition of HpEVs

In this study, *H. pluvialis* 192.80, obtained from the Sammlung von Algenkulturen Culture Collection of Algae at Gottingen University, was cultured in Bold Basal Medium (BBM) [52] and incubated under high light (350 µmol·m^−2^·s^−1^) [53] and sodium acetate (45 mM) stress conditions [54,55]. *H. pluvialis* cells were sampled after the high light and high sodium acetate treatment for 0, 9, and 48 h, representing three different growth phases including the green vegetative motile cell stage, the green nonmotile cell stage, and the red nonmotile cyst stage, respectively. The isolation and characterization of HpEVs were in accordance with our previous reports [27]. They were stored under −80 °C till proteomics analysis, and three separate cultivation media for each growth stage of *H. pluvialis* were used. Briefly, the liquid culture of *H. pluvialis* was centrifuged at 2000× *g* for 10 min and 10,000× *g* for 30 min to remove the microalga cells and debris, and then the supernatants were microfiltered and ultrafiltered using 0.22 µm Durapore PVDF membranes and 100 kD Amicon Ultra-15 Centrifugal Filters (Merck KGaA, Darmstadt, Germany) to concentrate HpEVs. The ultimate HpEV pellets were washed and collected by centrifugation at 100,000× *g* for 80 min. The isolated HpEVs were abbreviated to HpEVs-1, HpEVs-2, and HpEVs-3, corresponding to the life cycle phases of *H. pluvialis*.

### 4.2. Protein Identification and Quantification Using TMT

Tandem mass tag (TMT) labeling was used to identify proteins in three HpEVs with two biological replicates. Proteins with differential abundance among different HpEVs (i.e., HpEVs-1, HpEVs-2, and HpEVs-3) were also identified. Briefly, proteins were extracted from HpEVs using an Exosome Protein Extraction Kit (Biolaibo Biotechnology Co., Beijing, China) following the manufacturer’s protocol. The concentration of the extracted HPEV proteins was determined using a BCA Protein Assay Kit (Thermo Fisher Scientific, Waltham, MA, USA) according to the manufacturer’s instructions [56]. Before the proteomic analysis was performed, the quality of HpEV proteins (in two replicates, R1 and R2) was evaluated using 10% sodium dodecyl sulfate–polyacrylamide gel electrophoresis (SDS-PAGE).

Subsequently, the proteomic analysis of HpEVs was performed by Jingjie PTM BioLabs (Hangzhou, China). Before digestion, the protein solution was reduced with 5 mM dithiothreitol at 56 °C for 30 min, followed by alkylation with 11 mM iodoacetamide at room temperature for 15 min in dark conditions. Then, the protein samples were diluted using 100 mM TEAB to achieve a urea concentration of less than 2 M. Trypsin was then added to the protein samples at 1:50 (mass ratio) for the first overnight digestion. Trypsin was added again to the first digested solution at 1:100 (mass ratio) for another 4 h digestion. Then, the digested proteins (peptide mixture) were desalted using a C18 SPE column, labeled using TMT, fractionated using high-performance liquid chromatography, and characterized using liquid chromatography with tandem mass spectrometry (LC-MS/MS). Analyses were performed using MaxQuant’s default parameters with a false discovery rate of 1% at the protein and peptide level (v.1.6.6.0 http://www.maxquant.org/, accessed on 3 March 2020), and the database used was SwissProt (http://www.expasy.ch/sprot/, accessed on 3 March 2020). Enzyme specificity was considered fully cleaved by trypsin, with two maximum missed cleavage sites. The minimum peptide length required was seven amino acids.

### 4.3. Bioinformatics Analysis

To acquire insight into the proteomic profile of HpEVs, the identified proteins were screened in several databases and analyzed using bioinformatics methods, as described in previous reports [57,58,59,60]. Briefly, hierarchical clustering analysis was carried out using the R Package pheatmap (v.2.0.3 https://cran.r-project.org/web/packages/cluster/, accessed on 3 March 2020), and the heat map was produced by a dendrogram depicting the extent of similarity in the protein expression among samples. Gene Ontology (GO) proteome annotation was performed using the InterProScan database based on proteins that were classified into three categories, including biological process, cellular component, and molecular function (v.5.14-53.0 http://www.ebi.ac.uk/interpro/, accessed on 3 March 2020). The Orthologous Groups of Proteins (COGs) database (https://www.ncbi.nlm.nih.gov/research/cog-project/, accessed on 3 March 2020) and InterProScan software were used to perform the functional classification of identified proteins (v.5.14-53.0 http://www.ebi.ac.uk/interpro/, accessed on 3 March 2020). Wolfpsort and CELLO were used to predict the subcellular localization of the identified proteins (v.0.2 http://www.genscript.com/psort/wolf_psort.html, accessed on 3 March 2020; v.2.5 http://cello.life.nctu.edu.tw/, accessed on 3 March 2020). The KEGG database was used to identify enriched pathways by a two-tailed Fisher’s exact test (v.2.0 http://www.genome.jp/kaas-bin/kaas_main, accessed on 3 March 2020), and the KEGG Mapper website was used to classify enriched pathways into hierarchical categories (V2.5 http://www.kegg.jp/kegg/mapper.html, accessed on 3 March 2020). The Perl module was used to enrich the identified proteins (v.1.31 https://metacpan.org/pod/Text::NSP::Measures::2D::Fisher, accessed on 3 March 2020). The Blast database combined with the R package networkD3 (v.0.4 https://cran.r-project.org/web/packages/networkD3/, accessed on 3 March 2020) was used to construct protein–protein interaction networks, and the resulting networks were imported into Cytoscape v3.4.0. A *p*-value < 0.05 was controlled for significant enrichment. The mass spectrometry proteomic data were deposited to the ProteomeXchange Consortium via the PRIDE partner repository with the dataset identifier PXD051125.

### 4.4. Functional Analysis of Differentially Abundant Proteins

To analyze the differentially abundant proteins (DAPs), proteins in the HpEVs were quantified as the average from two biological replicates and were quantitatively compared between two groups (HpEVs-1 versus HpEVs-2, HpEVs-2 versus HpEVs-3, and HpEVs-1 versus HpEVs-3). The log2 intensity values were used to represent the relative quantitative values during the pairwise comparisons. *p*-values were calculated using the two-tailed *t*-test method to study the significant differences. The statistical analysis of DAPs was carried out using the R package Limma (version 3.83; www.bioconductor.org/packages/2.8/bioc/html/limma.html, accessed on 3 March 2020) [61]. A relative quantitative value exceeding 1.5 between two contrasts was considered a significant upregulation, while a value lower than 0.5 was considered a significant downregulation, with an adjusted *p*-value < 0.05. Then, bioinformatic analysis of the DAPs was performed as described above. The workflow for the proteomic analysis is illustrated in Figure 8. All the analyses were supported by Jingjie PTM Biolabs Co., Ltd. (Hangzhou, China).

## 5. Conclusions

The present work first examined the distinct proteomic composition characteristics of EVs derived from the cultivation medium of *H. pluvialis* in three growth phases. The comparative analysis extensively studied the distinct features among these three HpEVs. To summarize, *H. pluvialis* arranged parent-cell origin proteins into HpEVs with nature and functions coupled with cell status, thereby endowing these vesicles with a cell type-specific nature and biological functions. The homogeneity in HpEV protein components might also suggest the qualitative stability of HpEVs acquired from various cultivation conditions or life stages. The observations and suggestions made here may help us to understand the molecular basis of the function of EVs in cell-to-cell communication in microalgae. In addition, the results also established the potential functions of microalgae-derived EVs in trophic-level interactions and population substitution in natural environments, thereby deepening our understanding of the stability of the microalgal cultivation system. Furthermore, this understanding of the cargo of microalgae-derived EVs could further be exploited in clinical settings for both pharmacotherapy and targeted drug delivery. Overall, this study provides valuable insights into the functional properties of *H. pluvialis*-derived EVs and their potential applications in various fields.

## Figures and Tables

**Figure 1 ijms-25-05421-f001:**
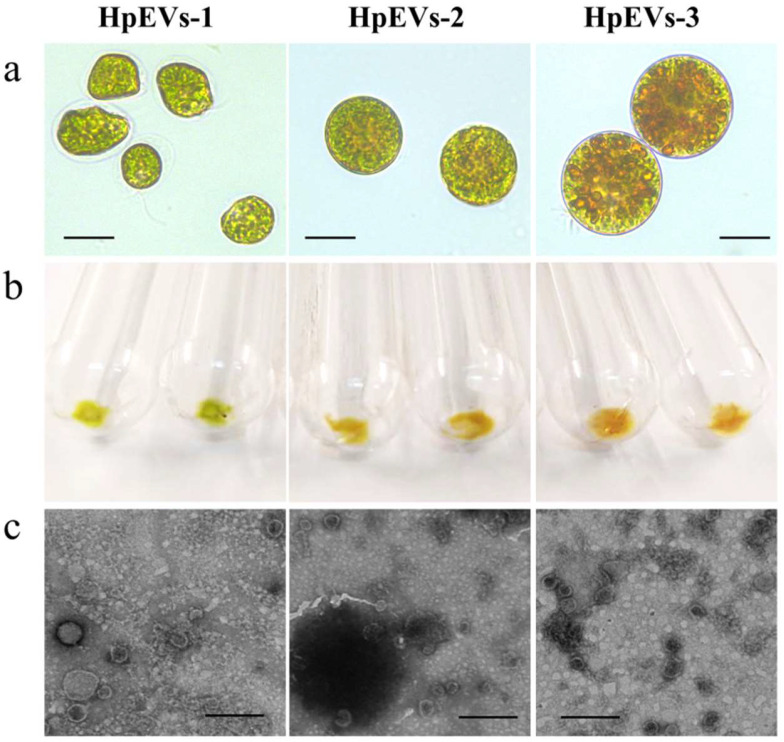
Characterization of *H. pluvialis*-derived extracellular vesicles (HpEVs). (**a**) Microscopic images of *H. pluvialis* cells in three growth phases (scale bar = 20 μm). (**b**) Output of the isolated HpEV pellets. (**c**) Transmission electron microscopy images of the HpEVs (scale bar = 200 nm).

**Figure 2 ijms-25-05421-f002:**
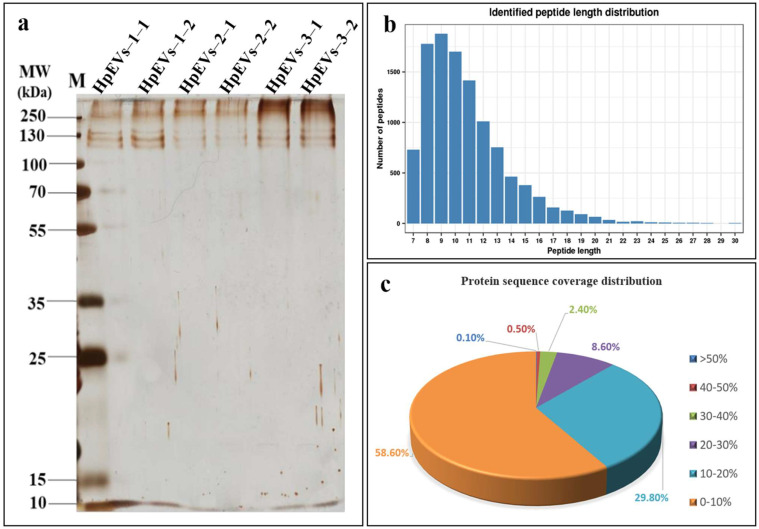
Characteristics of the identified HpEV peptides and proteins. (**a**) Total gel staining of the HpEV proteins. (**b**) The length distribution of the identified peptides in the identified HpEV proteins. (**c**) Sequence coverage distribution of the identified proteins.

**Figure 3 ijms-25-05421-f003:**
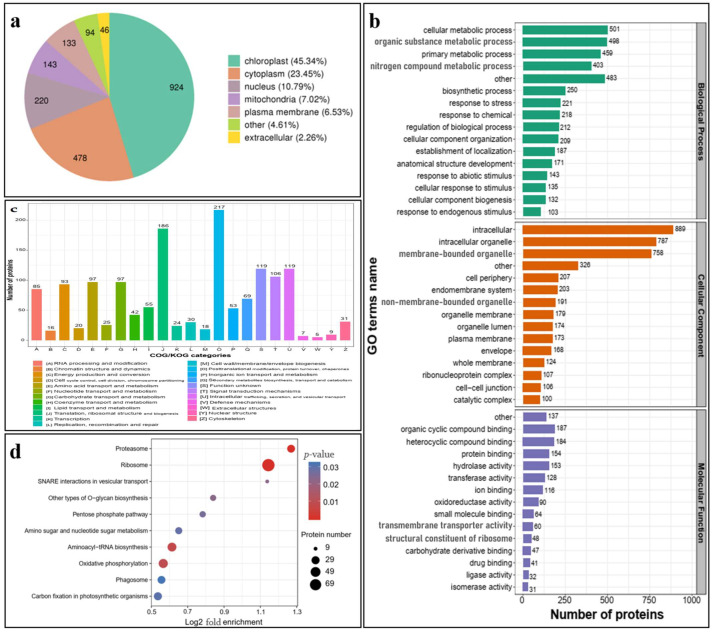
Analysis of the quantified HpEV proteins. (**a**) Subcellular localization of the quantified HpEV proteins. (**b**) Gene Ontology (GO) annotation of the HpEV proteome. (**c**) Clusters of orthologous groups (COG) functional classification of the HpVE proteome. (**d**) Kyoto Encyclopedia of Genes and Genomes (KEGG) enrichment analysis of the HpVE proteome.

**Figure 4 ijms-25-05421-f004:**
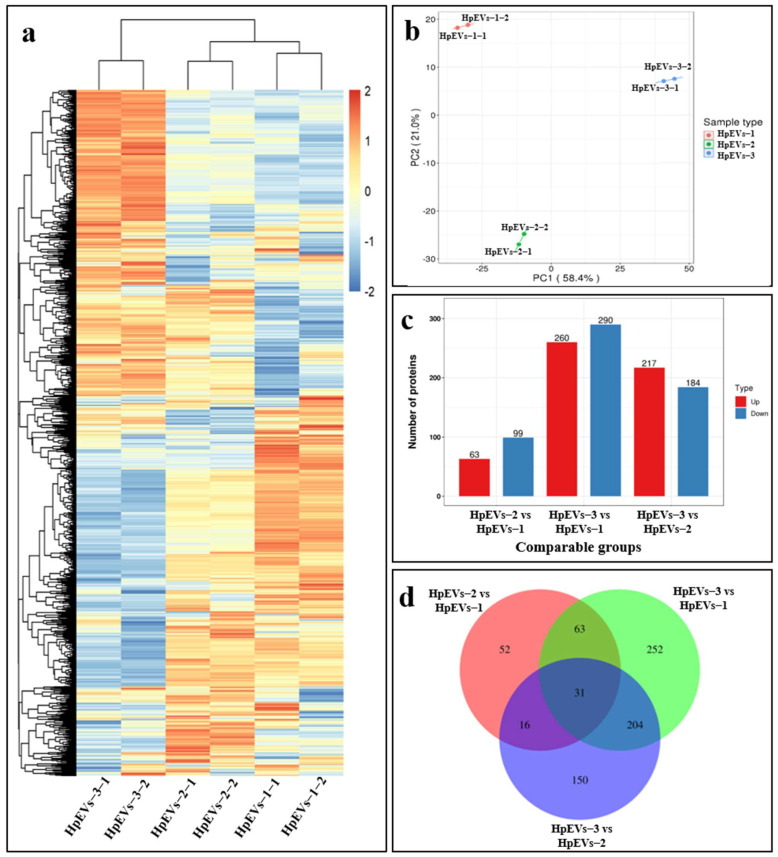
Quantified proteins and DAPs in HpEVs. (**a**) A clustering analysis was performed to display the significant DAPs in the three HpEVs. Red represents high abundance, while blue represents a relatively low abundance. (**b**) Principal component analysis (PCA) of proteins in the three HpEVs. (**c**) DAPs of the three HpEVs. (**d**) Venn diagram of the number of DAPs identified in the three HpEVs.

**Figure 5 ijms-25-05421-f005:**
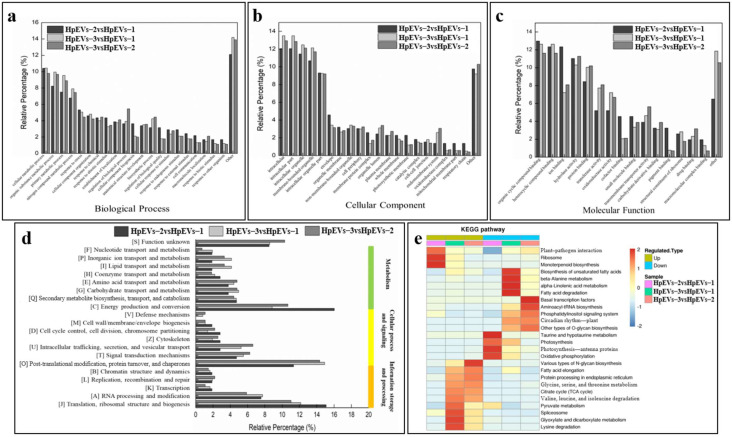
Bioinformatics analysis of the DAPs in HpEVs. (**a**–**c**) GO classification analysis of DAPs in different HpEVs to biological processes (**a**), cell components (**b**), and molecular functions (**c**) (*p*-value < 0.01). (**d**) COG classification of DAPs in different HpEVs. (**e**) KEGG pathway analysis of significantly enriched pathways from different comparison groups.

**Figure 6 ijms-25-05421-f006:**
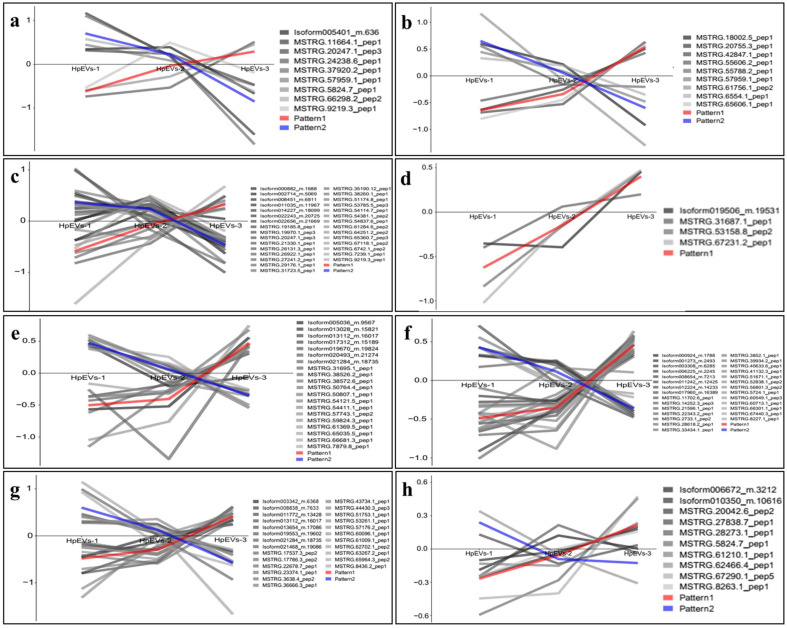
Abundance of DAPs in specific COG functional categories. (**a**–**h**) Abundance pattern of DAPs in the categories “cell cycle control, cell division, chromosome partitioning” (**a**), “replication, recombination and repair” (**b**), “signal transduction mechanisms” (**c**), “cell wall/membrane/envelope biogenesis” (**d**), “lipid transport and metabolism” (**e**), “intracellular trafficking, secretion, and vesicular transport” (**f**), “secondary metabolite biosynthesis, transport, and catabolism” (**g**), and “astaxanthin biosynthesis” (**h**).

**Figure 7 ijms-25-05421-f007:**
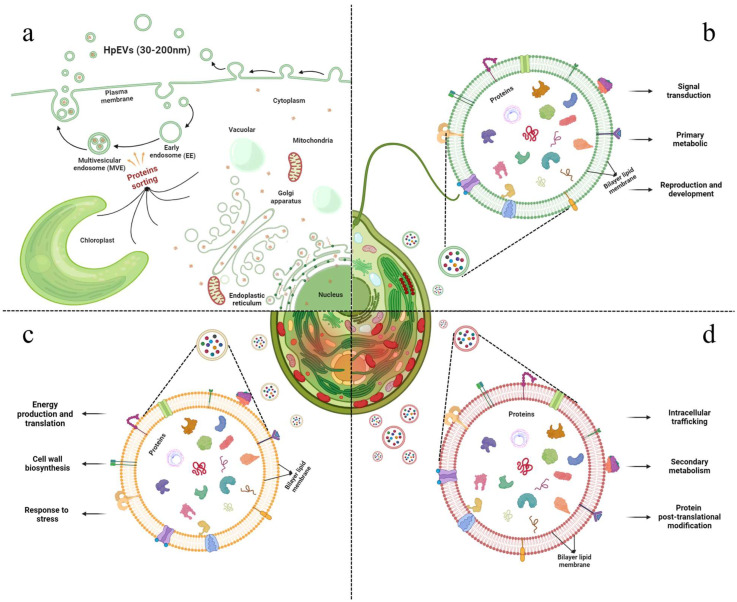
HpEVs produced in *H. pluvialis*. (**a**) The sketch mapping HpEV production and release. (**b**) The protein formation and predictive function of HpEVs-1. (**c**) The protein formation and predictive function of HpEVs-2. (**d**) The protein formation and predictive function of HpEVs-3.

**Figure 8 ijms-25-05421-f008:**
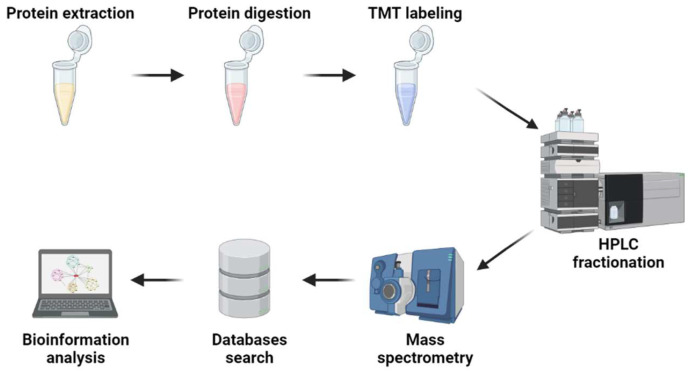
Workflow for the proteome analysis in this study. Arrows represent outputs.

**Table 1 ijms-25-05421-t001:** Protein concentration of the isolated HpEVs.

Treatment	Protein Concentration (μg·mL^−1^)
HpEVs-1	37.85 ± 2.03 ^c^
HpEVs-2	44.57 ± 4.44 ^b^
HpEVs-3	188.66 ± 22.87 ^a^

Note: Different lowercase letters indicate significant differences (*p* < 0.05) between two datasets and the same lowercase letter indicates the difference was not significant (*p* > 0.05).

## Data Availability

The original contributions presented in the study are included in the article/Appendix A, further inquiries can be directed to the corresponding author/s.

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
