# Peer review of "Comparative Proteome Profiling of Extracellular Vesicles from Three Growth Phases of Haematococcus pluvialis under High Light and Sodium Acetate Stresses"

_ijms, 2024, doi:10.3390/ijms25105421_

Round 1

Reviewer 1 Report

Comments and Suggestions for Authors

A well written paper.

However, I have few comments.

1. Include a flowchart to the methodology. 

2. Enhance your list of refences from the following.

doi.org/10.1016/j.jdermsci.2022.04.004

 doi: 10.1038/s41417-023-00627-w

doi: 10.1007/s00125-023-05992-7

3. Enhance the novelty and showcase the discussion related to above papers.

Comments on the Quality of English Language

Minor checks are needed. 

Author Response

Author's Notes to Reviewer

Dear reviewer,

Thank you for giving us an opportunity to revise our manuscript titled “Comparative proteome profiling of extracellular vesicles from three growth phases of Haematococcus pluvialis under high light-sodium acetate stress” (ijms-2976953) and to resubmit it to “International Journal of Molecular Sciences” for further consideration. And we also appreciate your insightful and constructive comments and advice, and we have carefully addressed these concerns and made a proper revision of the manuscript. These comments and suggestions have not only enabled us to provide a highly improved manuscript but also inspired us to conduct more in-depth studies on the function and exploitation of microalgal-derived extracellular vesicles in future works. We have already discussed in detail contents of this article and made the corresponding modifications, and we believe that this paper has been improved considerably. We hope that these revisions successfully address your concerns and requirements and that this manuscript could be accepted by the journal “International Journal of Molecular Sciences”.

The following issues are our replies to the editors's and reviewers’s comments and suggestions.

Note: In the response letter, all the reviewers’ comments/suggestions are shown in blue, all the changes are highlight in yellow, and all the responses are shown in black. In the revised manuscript, all the changes and additions are highlight in yellow. Looking forward to hearing from you soon.

Thanks for reviewing our article again.

Best wishes.

Yours sincerely

Authors

Response letter,

  1. Include a flowchart to the methodology.

Author response: Thanks to reviewer for making this valuable comment. In this study, the isolation and characterization of HpEVs were in accordance with our previously report, which had given detailed information and flowchart of the methodology. Therefore, briefly processes of the acquisition of HpEVs were added in Material and methods section of this paper, but detailed description of the methods was not provided here to short the article. Please check the revised manuscript.

  1. Enhance your list of refences from the following.

doi.org/10.1016/j.jdermsci.2022.04.004

doi: 10.1038/s41417-023-00627-w

doi: 10.1007/s00125-023-05992-7

Author response: Thank to reviewer for pointing this out. As suggested by the reviewer, we have checked manuscript and added the suggested refences to the manuscript in the Discussion section at line 396 to 399 and ,439 to 442 and the list of refences has been updated. We suppose these studies could give a deeper insight to EVs composition and their biology function. Please check the revised manuscript.

  1. Enhance the novelty and showcase the discussion related to above papers.

Author response: Thanks to reviewer for making this insightful comment. We agreed with the reviewer’s assessment, and enhanced the novelty and showcase the discussion based on the added papers. Please check the revised manuscript.

Reviewer 2 Report

Comments and Suggestions for Authors

Title

In the special issue about stress there is no need to title each article including stress.

Just high sodium acetate concentration is the best, whether it is stress only results may show.

English of the title is bad – with three stages – should be rephrased.

Abstract

“Haematococcus pluvialis-derived extracellular vesicles (HpEVs) were isolated from H. pluvialis cultivation medium” – too much Hp

high light and high sodium acetate stresses – „high“ should be quantified

Keywords – not valid due to title replication

Introduction

references 3-7 and 4, 8-13, 10-14, 19-21, 20, 22-23 should be splited inside sentences providing separately reference for each phenomenon mentioned. 

Systematic point is missing: microalgae as a novel natural source of EVs should be described mentioning main algae species, their systematic position.  No information about Hp selection opportunities compared to the other species

high sodium acetate (45 mM) stress conditions – no information is provided about low concentrations, no references provided how these high concentrations were selected and for what species it has been done, also treatment duration should have references. What is criteria for titling of that concentration as a stress?

The same true for „high light“ - (350 μmol·m–2·s–1) conditions – reference for selection of light intensities is missing.

what is criteria for titling of these concentrations as a stress?

Reference for Bold Basal Medium (BBM) suitability for Hp is missing.

Reference for Protein Assay Kit is missing.

References for statistical packages are missing.

the identified proteins were searched on databases –please, list the databases

„among different growth stages groups“ – English, style?

Content of Fig. 1 is too primitive.

Fig. 2 „growth stages“ without numbering – numbers are not explaned in the figure.

Fig. 3 characters are unclear and too small, size should be similar to the text words.

The characters in the Figures 4, 5 and 6, 7, and  8 in particular are too small and unreadable

It might be quicker understanding for the reader in case HpEV in the figures should be provided without abbreviation (abbreviation in the parantheses).

Figure 7 – logistic error „during the stage changes of HpEVs“.  which stages are – algae or vesicle

Discussion

The first paragraph of discussion – belongs to introduction

Bacterial EV are mentioned although comparisons with the other microalgae species are not provided

In the discussion references for the result tables and figures are nor provided.

Comments on the Quality of English Language

In many places Etext should be chacked by language specialist

Author Response

Author's Notes to Reviewer

Dear reviewer,

Thank you for giving us an opportunity to revise our manuscript titled “Comparative proteome profiling of extracellular vesicles from three growth phases of Haematococcus pluvialis under high light-sodium acetate stress” (ijms-2976953) and to resubmit it to “International Journal of Molecular Sciences” for further consideration. And we also appreciate your insightful and constructive comments and advice, and we have carefully addressed these concerns and made a proper revision of the manuscript. These comments and suggestions have not only enabled us to provide a highly improved manuscript but also inspired us to conduct more in-depth studies on the function and exploitation of microalgal-derived extracellular vesicles in future works. We have already discussed in detail contents of this article and made the corresponding modifications, and we believe that this paper has been improved considerably. We hope that these revisions successfully address your concerns and requirements and that this manuscript could be accepted by the journal “International Journal of Molecular Sciences”.

The following issues are our replies to the editors's and reviewers’s comments and suggestions.

Note: In the response letter, all the reviewers’ comments/suggestions are shown in blue, all the changes are highlight in yellow, and all the responses are shown in black. In the revised manuscript, all the changes and additions are highlight in yellow. Looking forward to hearing from you soon.

Thanks for reviewing our article again.

Best wishes.

Yours sincerely

Authors

Response letter,

Title

  1. In the special issue about stress there is no need to title each article including stress. Just high sodium acetate concentration is the best, whether it is stress only results may show. English of the title is bad – with three stages – should be rephrased.

Author response: Thanks to reviewer for making this valuable comment. We have checked the Title and rephrased it, the revised Title read as follows on “Comparative proteome profiling of extracellular vesicles from three growth phases of Haematococcus pluvialis under high light-sodium acetate stress”. We hope that the reviewer will satisfy the revised title.

Abstract

  1. Haematococcus pluvialis-derived extracellular vesicles (HpEVs) were isolated from H. pluvialis cultivation medium” – too much Hp

Author response: Thanks to reviewer for pointing this out. The reviewer is correct, and we have checked the manuscript and rewrote the Abstract in the revised manuscript. Please check the revised manuscript. 

  1. high light and high sodium acetate stresses – “high” should be quantified.

Author response: Thanks to reviewer for making this comment. As suggested by the reviewer, we have quantified the high light-sodium acetate stress in Abstract in the revised manuscript. Please check the revised manuscript.

  1. Keywords – not valid due to title replication

Author response: Thanks to reviewer for making this insightful comment. The reviewer is correct, we have checked the manuscript and rewrote the Keywords in the revised manuscript. The updated Keywords were Extracellular vesicles, Haematococcus pluvialis, Different growth phases, Proteome profile, and Biological functions. Please check the revised manuscript.

Introduction

  1. references 3-7 and 4, 8-13, 10-14, 19-21, 20, 22-23 should be splited inside sentences providing separately reference for each phenomenon mentioned. 

Author response: Thanks to reviewer for making this valuable comment. We agreed with the point of the reviewer, and checked throughout the manuscript to splite these references inside sentences to provide separately reference for each phenomenon mentioned. Please check the revised manuscript.

  1. Systematic point is missing: microalgae as a novel natural source of EVs should be described mentioning main algae species, their systematic position.  No information about Hp selection opportunities compared to the other species.

Author response: Thanks to reviewer for making this insightful comment. We agreed with the point of the reviewer, and checked the whole text to coordinate this article and organize our ideas logically. In the Introduction section, we added systematic points about microalgae for natural source of EVs, and information about Haematococcus pluvialis selection opportunities compared to the other microalgal species. Please check the revised manuscript.

Material and methods

  1. high sodium acetate (45 mM) stress conditions – no information is provided about low concentrations, no references provided how these high concentrations were selected and for what species it has been done, also treatment duration should have references. What is criteria for titling of that concentration as a stress?

Author response: Thanks to reviewer for making this insightful comment. For the microalgae H. pluvialis, many previous studies have shown that sodium acetate accelerates its cell growth or astaxanthin accumulation, and exogenous sodium acetate with concentration of < 10mM enhanced astaxanthin accumulation and photoprotection capacity of this microalgae [Orosa et al., 2001, Zhang et al., 2019]. It was reported that 45 mM sodium acetate was identified as an environmental stress for H. pluvialis to accumulate large amounts of astaxanthin as results of the up-regulation of related genes. And, it was also reported by Wang et al [2021] that the final concentration of sodium acetate was 45 mM for the sodium acetate stress (HS), which was considered as high sodium acetate stress. Hence, we selected the concentration of 45 mM as high sodium acetate (45 mM) stress conditions in this study. And, we have added the suggested reference to the manuscript for selection of high sodium acetate stress. Please check the revised manuscript.

REFERENCES

Orosa M, Franqueira D, Cid A, Albade J. Carotenoid accumulation in Haematococcus pluvialis in mixotrophic growth. Biotechnology Letters 2001, 23:373–378.

Wang CG, Wang KP, Ning JJ, Luo QL, Yang Y, Huang DQ, Li H. Transcription factors from Haematococcus pluvialis involved in the regulation of astaxanthin biosynthesis under high light-sodium acetate stress. Frontiers in Bioengineering and Biotechnology 2021, 9(650178): 1–12.

Zhang CH, Zhang LT, Liu JG, 2019. Exogenous sodium acetate enhances astaxanthin accumulation and photoprotection in Haematococcus pluvialis at the non-motile stage. Journal of Applied Phycology 2019, 31:1001–1008.

  1. The same true for “high light” - (350 μmol·m–2·s–1) conditions – reference for selection of light intensities is missing. What is criteria for titling of these concentrations as a stress?

Author response: Thanks to reviewer for making this comment. The green algae stain studied in this manuscript was H. pluvialis 192.80, which was obtained from Sammlung von Algenkulturen Gottingen Culture Collection of Algae, and was conserved in our lab under continuous fluorescent light with intensity of 20 μmol·m-2·s-1. In many published reports, light intensity used for vegetative cultivation of this microalga was 20 μmol·m–2·s–1[Allewaert et al., 2015; Zheng et al., 2017; Luo et al., 2017, 2018], and the light intensity used for genome sequencing of this microalgae, which was used to quantify the transcriptome of H. pluvialis, was also 20 μmol·m–2·s–1 [Luo et al., 2018]. It was previously reported that light intensities > 300 µmol photons m-2·s-1 increased the dry biomass and astaxanthin accumulation of autotrophic H. pluvialis, and the dry biomass productivity and astaxanthin accumulation were not significantly different between a light intensity range of 300–1000 µmol photons m-2·s-1. And, it was previously defined by Wang et al [2014] that the high light (HL) illumination (µmol photons m-2·s-1) condition was used for the cultivation of H. pluvialis across life cycle stages. Additionally, strong light intensity of 350 µmol photons m-2·s-1 was employed to induce the accumulation of astaxanthin in H. pluvialis [Orosa et al., 2001; Do et al., 2019]. Synthesize the above results into consideration, we chose the light intensity of 350 µmol photons m-2·s-1 as stress. And, we have added the suggested reference to the manuscript for selection of high light intensities. Please check the revised manuscript.

REFERENCES

Allewaert CC, Vanormelingen P, Proschold T, Gomez PI, Gonzalez MA, Bilcke G, D’Hondt S, Vyverman W. Species diversity in European Haematococcus pluvialis (Chlorophyceae, Volvocales). Phycologia 2015, 54 (6): 583–598

Do TT, Ong BN, Tran MLN, Nguyen D, Melkonian M, Tran HD. Biomass and astaxanthin productivities of Haematococcus pluvialis in an angled twin-layer porous substrate photobioreactor: effect of inoculum density and storage time. Biology 2019, 8(68): 1–14.

Luo Q L, Bian C, Tao M, Huang Y, Zheng YH, Lv YY, Li J, Yang CG, You XX, Jia B, Xu JM, Li JC, Li Z, Shi Q, Hu ZL. Genome and transcriptome sequencing of the astaxanthin-producing green microalga, Haematococcus pluvialis. Genome Biology and Evolution 2018, 11: 166–173

Luo QL, Wang KP, Xiao K, Wang CG, Hu ZL. A rapid and high-quality method for total RNA isolation from Haematococcus pluvialis. Genetics and Molecular Research 2017, 16 (2): gmr16029614

Orosa M, Valero JF, Herrero C, Abalde J. Comparison of the accumulation of astaxanthin in Haematococcus pluvialis and other green microalgae under N-starvation and high light conditions. Biotechnology Letters 2001, 23: 1079-1085.

Wang B, Zhang Z, Hu Q, Sommerfeld M, Lu Y, Han D. Cellular capacities for high-light acclimation and changing lipid profiles across life cycle stages of the green alga Haematococcus pluvialisPloS one 2014, 9(9): e106679.

Zheng YH, Li Z, Tao M, Li JC, Hu ZL. Effects of selenite on green microalga Haematococcus pluvialis: Bioaccumulation of selenium and enhancement of astaxanthin production. Aquatic Toxicology 2017, 183: 21–27

  1. Reference for Bold Basal Medium (BBM) suitability for Hp is missing. Reference for Protein Assay Kit is missing. References for statistical packages are missing.

Author response: Thank to reviewer for pointing this out. As suggested by the reviewer, we have checked manuscript and added the related refences and website to the manuscript, and enhanced our list of refences. Please check the revised manuscript.

  1. the identified proteins were searched on databases –please, list the databases

Author response: Thank to reviewer for pointing this out. We have checked manuscript and added the database for the identification of proteins, and the other related software or bioinformatics analysis tools were also provided in the revised manuscript. Please check the revised manuscript.

  1. “among different growth stages groups” – English, style?

Author response: Thanks to reviewer for making this insightful comment. As well known, the life cycle of H. pluvialis consists of four types of distinguishable cellular morphologies: macrozooids (zoospores), microzooids, palmella, and hematocysts (aplanospores) [Hazen, 1899; Elliot, 1934; Shah et al., 2016]. These four types of H. pluvialis were also described in previously reports as growth stages or phases [Shah et al., 2016; Sun et al., 2017; Reinecke et al., 2018]. As the reviewer raised this question, we searched the published reports again, and found that “growth phase” may more suitable to describe the H. pluvialis used for EVs isolation in this study. Hence, we check throughout the manuscript and rewrote all these words. Please check the revised manuscript.

REFERENCES

Elliot AM. Morphology and life history of Haematococcus pluvialis. Arch. Protistenk. 1934, 82: 250–272.

Hazen TE. The life history of Sphaerella lacustris. Mem. Torrey Bot. Club 1899, 6: 211–244.

Reinecke DL, Castillo-Flores A, Boussiba S, Zarka A. Polyploid polynuclear consecutive cell-cycle enables large genome-size in Haematococcus pluvialis. Algal Research 2018, 33: 456–461.

Shah MMR, Liang Y, Cheng JJ, Daroch M. Astaxanthin-producing green microalga Haematococcus pluvialis: from single cell to high value commercial products. Frontiers in Plant Science 2016, 7(531): 1–28.

Sun H, Liu B, Lu X, Chen KW, Chen F. Staged cultivation enhances biomass accumulation in the green growth phase of Haematococcus pluvialis. Bioresource Technology 2017, 233: 326–331.

  1. Content of Fig. 1 is too primitive.

Author response: Thanks to reviewer for making this comment. We agreed with the reviewer that the content of Figure 1 is primitive, but the purpose of this figure was to provide a flowchart to the methodology used for proteome analysis of HpEVs. In consideration of the time limitation for revision and content in Figure 1 have represented the realistic methods and processes of this study, we didn’t update this figure. Thanks again for your kind suggestion.

Results

  1. Fig. 2 “growth stages” without numbering – numbers are not explaned in the figure.

Author response: Thanks to reviewer for making this comment. We have checked this figure, and found that the number of HpEVs, abbreviated as HpEVs-1, HpEVs-2, and HpEVs-3, were explained at the top of the figure in the “3.1 Obtainment of HpEVs from H. pluvialis culture medium under high light and high sodium acetate stresses” section. Please check the revised manuscript.

Figure 2 | Characterization of H. pluvialis derived extracellular vesicles (HpEVs). (a) Microscopic images of H. pluvialis cells in three growth phases (scale bar = 20 μm). (b) Output of the isolated HpEVs pellets. (c) Transmission electron microscopy images of the HpEVs (scale bar = 200 nm).

  1. Fig. 3 characters are unclear and too small, size should be similar to the text words. The characters in the Figures 4, 5 and 6, 7, and 8 in particular are too small and unreadable.

Author response: Thanks to reviewer for this valuable comment. We checked these figures, and tried our best to enlarge the font size in all these figures without compromising the quality of them. In addition, we also provided figures with higher resolution to make it convenient for the readers to magnify the picture for detail view. Please check the revised manuscript.

  1. It might be quicker understanding for the reader in case HpEV in the figures should be provided without abbreviation (abbreviation in the parantheses).

Author response: Thanks to reviewer for this comment. We agreed with the reviewer’s assessment. As suggested by the reviewer, we have updated thess figures and replaced all the abbreviation of EV into HpEV as requested.

  1. Figure 7 – logistic error “during the stage changes of HpEVs”.  which stages are-algae or vesicle

Author response: Thanks to reviewer for making this comment. The “stage” mentioned “during the stage changes of HpEVs” was the phase of vesicles. We have checked the figure note and detailed descript this figure as “Figure 7 | Abundance of DAPs in specific COG functional categories”. And, we checked throughout the manuscript and updated the related sentences to dispel misunderstanding. Please check the revised manuscript.

Discussion

  1. The first paragraph of discussion – belongs to introduction

Author response: Thanks to reviewer for this valuable suggestion. In order to take this concern into account, and improve the quality of our manuscript, the content in the first paragraph of Discussion was changed, and the existing content was transferred to the Introduction section.

The updated first paragraph of Discussion was as follows:

It is generally believed that extracellular vesicles (EVs) are important mediators of intercellular communication via transfering a wide variety of molecular cargoes [1,8,13,42-43]. The property of EVs that carry parent-cell-specific signatures permits target-cell interactions in distinctly different manners [44]. Understanding of this complexity and addressing its characterization and production processes is essential for the drug delivery perspective [11]. The protein components in EVs not only reflect the specific parent-cell origins, but also are vary depending on the nature and biogenesis of different EV subpopulations [45]. Thus, attempts to analyse the protein cargo of EVs are essential for the understanding of their cellular functions and biological properties [30], and are also important in further researches regarding their utilization [15]. The exploitation of H. pluvialis derived EVs (HpEVs) could be considered as tailor-made high-value-added bioproducts of astaxanthin or biomass, which maybe a potential field of microalgae biotechnology [21,23-24].

And, the related updated content of Introduction was as follows:

The unicellular freshwater microalga Haematococcus pluvialis is considered as the best natural source for astaxanthin production for a long time [28]. Moreover, H. pluvialis also simultaneously accumulates diverse metabolites with high commercial value during astaxanthin accumulation in the red stage, including lipids, carbohydrates, and proteins, making it an attractive feedstock for multiple-product biorefining toward a higher commercial realization [28-30]. Therefore, H. pluvialis-derived EVs (HpEVs) can also be carried out through the integration as a novel production process, because HpEVs can be harvested from the culture media of H. pluvialis, which will not effect the biomass. The potential exploitation of HpEVs, as tailor-made high-value-added bioproducts of H. pluvialis biomass, will be a noteworthy attempt for future development of H. pluvialis biotechnology. For this, the fundamental knowledge of HpEVs’ physiology, including their morphology and function in microalgal growth and metabolism, must be initially determined. Understanding the composition of HpEVs is the key to understand their contributions to cellular and molecular regulation in H. pluvialis. However, to our best knowledge, the proteomics basis of HpEVs remains unknown. Proteomic analysis has been widely used to characterize various types of protein cargoes in EVs [1,11,31]. In this study, proteomic analyses were performed to characterize the protein cargoes of EVs derived from H. pluvialis at different growth phases, including the green vegetative motile cell stage, green nonmotile cell stage, and red nonmotile cyst stage, using a high-resolution liquid chromatography-tandem mass spectrometry (LC-MS/MS)-based protein quantification approach with isobaric multiplex tandem mass tag (TMTs). Classification of the subcellular localization of the quantified proteins and proteinrelated biological processes and metabolic pathways is necessary for the identification of specific features of HpEVs based on bioinformatics analysis. Furthermore, comparative research on these proteomes will unveil the distinctive functions of differential HpEVs along with the growth and astaxanthin accumulation in H. pluvialis. This study provide novel insights into the protein composition of HpEVs via proteomic strategies and depict the potential mechanism of cell-to-cell communication, results will provide a basis for further research on the potential application of HpEVs as high value coproducts of H. pluvialis biomass.

Please check the revised manuscript.

  1. In the discussion, references for the result tables and figures are nor provided.

Author response: Thanks to reviewer for point this out. We have carefully addressed this concern, and added and provided references for the result tables and figures in the Discussion section. Please check the revised manuscript.

  1. Comments on the Quality of English Language. In many places Etext should be chacked by language specialist

Author response: Thanks for this suggestion, the writing skill has been improved. We have discussed the writing problems with a professional English supervisor who graduated from North Dakota State University and worked in USA for 10 years. With her help, we made a lot of changes and revised our errors. We feel sorry for causing you unnecessary troubles in reviewing our manuscript, we hope that the revised version would satisfy you.

Reviewer 3 Report

Comments and Suggestions for Authors

The authors have published recently a paper in Biotechnology for Biofuels and Bioproducts 2024, 17(15): 1-20. https://doi.org/10.1186/s13068-024-02462-z, in which the  isolation, characterization, and functional verification of Haematococcus pluvialis-derived extracellular vesicles (HpEVs)  has been described along with small RNA high-throughput sequencing to determine the miRNA cargo of HpEVs. The present work is focused on the proteomic characterization of HpEVs, applying the same treatment to induce transitions between green vegetative motile stage, green nonmotile stage and red cyst stage. Having in view the scarcity of data about the molecular characteristics of HpEVs, their importance for cell-to-cell communication as membrane-enclosed vesicles released by cells carrying proteins, nucleic acids and metabolites, which can be transferred to a recipient cell to elicit a functional response, and the possibilities to use EVs as cargo delivering nanocarrier particles for bioactive molecules and/or artificial drug molecules, the present work contains valuable new information.

My remarks concern mainly the language and some methods description.

Introduction - Line 31 –“involving in cellular homeostasis compensating for the stress conditions” – unclear; line 59 – can be uptake, line 85 – differently abundant

MMs – how the different growth stages of H. pluvialis were defined –by time of growing , by microscopic appearance, by astaxantin content?  The previous published work on HpEVs could be cited for explanation. How the purity of CV preparations was checked? The extraction was performed with Exosome Protein Extraction Kit – were there any controls for purity?

 Lines 108-118 – please check the trypsin digestion protocol – usually reduction/alkylation with DTT/IAA follows trypsin digestion. Line 111 – “The digested protein samples were then diluted” and line 113 – “Finally, trypsin was added at a 1:50 trypsin-to-protein mass ratio for the first digestion” – if protein samples were already digested why trypsin was added for first digestion?

Line 138 – Why only two repeats of each group, there should be three independent replicates?

Results – Lines 181-182 “The mass of the most identified proteins ranged between 10 and 60 kDa” but fig 3a- SDS electrophoresis on 10% separating gel – majority of bands is above 100 kDa – how this discrepancy could be explained?

A question – what is the difference in EV proteome compared to the cellular proteome of the respective growth stage? Are EVs simply “buds” of the cells or they are expected to be enriched with some specific cargo proteins? It will be interesting to see how the reported EV proteome is related to the cellular proteome of the respective growth stage.

Comments on the Quality of English Language

The English of the manuscript needs special attention, particularly the verb forms and tenses. For example

Abstract lines 9-11 “However, utilization of these EVs is challenging due to their molecular composition remain poorly understood, resolvent of their composition could provide novel insights into biological properties and application characteristics researches” – understandable but should be edited in proper English (composition which remains, resolution/unraveling instead of resolvent, application characteristics researches – unclear, the last word is unnecessary. Line 16 – various instead of variant; lines 17-18 – “Thereby endowing HpEVs with early growth stage contain more proteins associated with cellular functions of primary metabolite” – “endowing” is unnecessary, HpEVs at early growth stage,  cellular functions of primary metabolite – unclear; line 20 – “provides the promising role” – unclear, maybe – proves instead of provides

Author Response

Author's Notes to Reviewer

Dear reviewer,

Thank you for giving us an opportunity to revise our manuscript titled “Comparative proteome profiling of extracellular vesicles from three growth phases of Haematococcus pluvialis under high light-sodium acetate stress” (ijms-2976953) and to resubmit it to “International Journal of Molecular Sciences” for further consideration. And we also appreciate your insightful and constructive comments and advice, and we have carefully addressed these concerns and made a proper revision of the manuscript. These comments and suggestions have not only enabled us to provide a highly improved manuscript but also inspired us to conduct more in-depth studies on the function and exploitation of microalgal-derived extracellular vesicles in future works. We have already discussed in detail contents of this article and made the corresponding modifications, and we believe that this paper has been improved considerably. We hope that these revisions successfully address your concerns and requirements and that this manuscript could be accepted by the journal “International Journal of Molecular Sciences”.

The following issues are our replies to the editors's and reviewers’s comments and suggestions.

Note: In the response letter, all the reviewers’ comments/suggestions are shown in blue, all the changes are highlight in yellow, and all the responses are shown in black. In the revised manuscript, all the changes and additions are highlight in yellow. Looking forward to hearing from you soon.

Thanks for reviewing our article again.

Best wishes.

Yours sincerely

Authors

Response letter,

The authors have published recently a paper in Biotechnology for Biofuels and Bioproducts 2024, 17(15): 1-20. https://doi.org/10.1186/s13068-024-02462-z, in which the  isolation, characterization, and functional verification of Haematococcus pluvialis-derived extracellular vesicles (HpEVs)  has been described along with small RNA high-throughput sequencing to determine the miRNA cargo of HpEVs. The present work is focused on the proteomic characterization of HpEVs, applying the same treatment to induce transitions between green vegetative motile stage, green nonmotile stage and red cyst stage. Having in view the scarcity of data about the molecular characteristics of HpEVs, their importance for cell-to-cell communication as membrane-enclosed vesicles released by cells carrying proteins, nucleic acids and metabolites, which can be transferred to a recipient cell to elicit a functional response, and the possibilities to use EVs as cargo delivering nanocarrier particles for bioactive molecules and/or artificial drug molecules, the present work contains valuable new information.

Author response: We are deeply grateful to the reviewer for this recommendation. The reviewer gives an accurate summary of our work and brings forward constructive questions, which not only enabled us to provide a highly improved manuscript but also inspired us to conduct more in-depth studies on the function and exploitation of microalgal-derived extracellular vesicles in future works. And we have carefully addressed the reviewer’s concerns and made a proper revision of the manuscript. We sincerely hope that this revised manuscript has addressed all your comments and suggestions.

My remarks concern mainly the language and some methods description.

Introduction

  1. Line 31 –“involving in cellular homeostasis compensating for the stress conditions” – unclear;

Author response: Thanks to reviewer for making this comment. We checked this sentence and rewrote as “EVs are also involved in maintaining cellular homeostasis and altering their metabolism to compensate cells’ respond to stress [2,3],” in line 31 to 32 in the revised manuscript. Please check the revised manuscript.

  1. line 59 – can be uptake.

Author response: Thanks to reviewer for making this comment. We apologize for the careless mistake. We checked this sentence and rewrote as “can be uptaken” in line 67 in the revised manuscript. Please check the revised manuscript.

  1. line 85 – differently abundant MMs – how the different growth stages of pluvialis were defined –by time of growing, by microscopic appearance, by astaxantin content?  The previous published work on HpEVs could be cited for explanation.

Author response: Thanks to reviewer for this valuable comment. We checked the manuscript, and the different growth stages of H. pluvialis were defined in the Material and methods section as green vegetative motile stage, green nonmotile stage, and red nonmotile cyst stage of the microalgal life cycle, which were identified by microscopic appearance as previously defined [Kobayashi et al., 1997; Hagen et al., 2022; Shah et al., 2016; Reinecke et al., 2018]. In which, the green vegetative motile cells are spherical, ellipsoidal, or pear-shaped cells with two flagella of equal length emerging from anterior end, and a cup-shaped chloroplast with numerous, scattered pyrenoids. The green nonmotile cells are green vegetative motile cells losing flagella and expanding their cell size, they form an amorphous multilayered structure in the inner regions of the extracellular matrix and become resting vegetative cells. The red nonmotile cysts are cells containing a thick sheath and cell wall, and accumulating large amounts of astaxanthin in lipid droplets deposited in the cytoplasm, with a characteristic bright red color. And, we also have defined this in our previous work [Hu et al., 2024].

REFERENCES

Hagen C, Siegmund S, Braune W. Ultrastructural and chemical changes in the cell wall of Haematococcus pluvialis (Volvocales, Chlorophyta) during aplanospore formation. European Journal of Phycology 2002, 37: 217–226.

Hu QJ, Hu ZL, Yan XJ, Lu J, Wang CG. Extracellular vesicles involved in growth regulation and metabolic modulation in Haematococcus pluvialis. Biotechnology for Biofuels and Bioproducts 2024, 17(15): 1–20.

Kobayashi M, Kurimura Y, Kakizono T, Nishio N, Tsuji Y. Morphological changes in the life cycle of the green alga Haematococcus pluvialis. Journal of Fermentation & Bioengineering 1997, 84(1): 94–97.

Reinecke DL, Castillo-Flores A, Boussiba S, Zarka A. Polyploid polynuclear consecutive cell-cycle enables large genome-size in Haematococcus pluvialis. Algal Research 2018, 33: 456–461.

Shah MMR, Liang Y, Cheng JJ, Daroch M. Astaxanthin-producing green microalga Haematococcus pluvialis: from single cell to high value commercial products. Frontiers in Plant Science 2016, 7(531): 1–28.

  1. How the purity of EV preparations was checked? The extraction was performed with Exosome Protein Extraction Kit – were there any controls for purity?

Author response: Thanks to reviewer for this insightful comment. In this study, the isolation of HpEVs contained the steps of ultrafiltration and washing, which removed the unwanted extracellular component, including most of the proteins. The extraction methods used in this study was consistent with the protocols previously reported involving in microalgal-derived EVs [Adamo et al., 2021; Paterna et al., 2022; Picciotto et al., 2021]. And the further characterization of HpEVs based on size and morphology has shown that the isolated HpEVs are purified at some level in our previous report [Hu et al., 2024]. Therefore, we believe that the extracted HpEVs were checked purity. Then, the quality of HpEV proteins was determined before the proteomic analysis to avoid significant differences among HpEVs caused by possible contamination of cells debris or proteins. We also think that the extracted proteins were purity proteins of HpEVs.

REFERENCES

Adamo G, Fierli D, Romancino DP, Picciotto S, Barone ME, Aranyos A, Božic D, Morsbach S, Raccosta S, Stanly C, Paganini C, Gai MY, Cusimano A, Martorana V, Noto R, Carrotta R, Librizzi F, Randazzo L, Parkes R, Palmiero UC, Rao E, Paterna A, Santonicola P, Iglic A, Corcuera L, Kisslinger A, Schiavi ED, Liguori GI, Liguori K, Kralj-Iglic V, Arosio P, Pocsfalvi G, Touzet N, Manno M, Bongiovanni A. Nanoalgosomes: introducing extracellular vesicles produced by microalgae. Journal of Extracellular Vesicles 2021, 10(e12081): 1–22.

Hu QJ, Hu ZL, Yan XJ, Lu J, Wang CG. Extracellular vesicles involved in growth regulation and metabolic modulation in Haematococcus pluvialis. Biotechnology for Biofuels and Bioproducts 2024, 17(15): 1–20.

Paterna A, Rao E, Adamo G, Raccosta S, Picciotto S, Romancino D, Noto R, Touzet N, Bongiovanni A, Manno M. Isolation of extracellular vesicles from microalgae: a renewable and scalable bioprocess. Frontiers in Bioengineering and Biotechnology 2022, 10(836747): 1–12.

Picciotto S, Barone ME, Fierli D, Aranyos A, Adamo G, Božič D, Romancino DP, Stanly C, Parkes R, Morsbach S, Raccosta S, Paganini C, Cusimano A, Martorana V, Noto R, Carrotta R, Librizzi F, Palmiero UC, Santonicola P, Iglič A, Gai M, Corcuera L, Kisslinger A, Schiavi ED, Landfester K, Liguori GL, Kralj-Iglič V, Arosio P, Pocsfalvi G, Manno M, Touzet N, Bongiovanni A. Isolation of extracellular vesicles from microalgae: towards the production of sustainable and natural nanocarriers of bioactive compounds. Biomaterials Science 2021, 9: 2917–2930.

  1. Lines 108-118 – please check the trypsin digestion protocol – usually reduction/alkylation with DTT/IAA follows trypsin digestion.

Author response: Thanks to reviewer for this suggestion. The checked the previous reports that involve in EVs/exosomes’ or H. pluvialis proteomic analysis, and the trypsin digestion protocol was widely used for LC-MS/MS analysis and Label-free/TMT quantification, the reduction/alkylation with DTT/IAA were not mentioned [Wang et al., 2022; Wang et al., 2022; Lee at al., 2023]. In this study, the procedures for HpEVs proteomic analysis were guided by Jingjie PTM BioLab (Hangzhou) Co. Ltd., the protocol related to reduction/alkylation with DTT/IAA were also not carried out [Forget et al., 2018; Liu et al., 2021; Gao at al., 2022; Yin et al., 2022]. Hence, the protocol of reducting/alkylating the digested protein of HpEVs with DTT/IAA was not contained in this study.

REFERENCES

Forget A, Martignetti L, Puget S, Calzone L, Brabetz S, Picard D, Montagud A, Liva S, Sta A, Dingli F, Arras G, Rivera J, Loew D, Besnard A, Lacombe J, Pages M, Varlet P, Dufour C, Yu H, Mercier AL, Indersie E. Aberrant ERBB4-SRC signaling as a hallmark of group 4 medulloblastoma revealed by integrative phosphoproteomic profiling. Cancer Cell 2018, 34: 379–395.

Gao JB, Pan T, Chen XL, Wei Q, Xu LY. Proteomic analysis of Masson pine with high resistance to pine wood nematodes. PLoS One 2022, 17(8): e0273010: 1–14.

Lee BH, Chen YZ, Shen TL, Pan TM, Hsu W H. Proteomic characterization of extracellular vesicles derived from lactic acid bacteria. Food Chemistry 2023, 427(136685): 1–10.

Liu WW, Zhou Y, Duan WZ, Song J, Wei S, Xia SK, Wang YY, Du XH, Li EC, Ren CX, Wang W, Zhan QM, Wang Q. Glutathione peroxidase 4-dependent glutathione high-consumption drives acquired platinum chemoresistance in lung cancer-derived brain metastasis. Clinical Translational Medicine 2021, e517: 1–22.

Wang XD, Meng CX, Zhang H, Xing W, Cao K, Zhu BK, Zhang CS, Sun FJ, Gao ZQ. Transcriptomic and proteomic characterizations of the molecular response to blue light and salicylic acid in Haematococcus pluvialis. Marine drugs 2022, (20,1): 1–25.

Wang BB, Pan XH, Wang F, Liu LL, Jia J. Photoprotective carbon redistribution in mixotrophic Haematococcus pluvialis under high light stress. Bioresource Technology 2022, 362(127761): 1–11.

Yin XJ, Li M, Wang YZ, Zhao GF, Yang T, Zhang YQ, Guo JB, Meng TT, Du RL, Li HL, Zhang J, He QY. Herbal medicine formula Huazhuo Tiaozhi granule ameliorates dyslipidaemia via regulating histone lactylation and miR-155-5p biogenesis. Clinical Epigenetics 2023, 15(175): 1–14.

  1. Line 111 – “The digested protein samples were then diluted” and line 113 – “Finally, trypsin was added at a 1:50 trypsin-to-protein mass ratio for the first digestion” – if protein samples were already digested why trypsin was added for first digestion?

Author response: Thanks to reviewer for this comment. We checked the manuscript and confirmed that this was an error, we apologize for the careless mistake. We rewrote these sentences as “Subsequently, the proteomic analysis of HpEVs were performed by Jingjie PTM BioLabs (Hangzhou, China). Before digestion, the protein solution was reduced with 5 mM dithiothreitol at 56 °C for 30 min, followed by alkylation with 11 mM iodoacetamide at room temperature for 15 min in dark condition. Then the protein samples were diluted using 100 mM TEAB to achieve a urea concentration less than 2M. Trypsin was then added into protein samples at 1:50 (mass ratio) for the first overnight digestion. Trypsin was added again to the first digested solution at 1:100 (mass ratio) for another 4 h-digestion.” in line 123 to 128. Please check the revised manuscript.

  1. Line 138 – Why only two repeats of each group, there should be three independent replicates?

Author response: Thanks to reviewer for this valuable comments. As we known, H. pluvialis is unicellular fresh water microalga, and the suspension culture contained large amounts of individual cells. Thus, the isolated HpEVs in this study were a complex of EVs secreted by H. pluvialis with consistent cell states, which were accumulated in the culture media with volume of about 1 liter. According to previous reports related to proteomic analysis, proteomic studies frequently have only one or two replicates, and the analytical method were used for statistical tests for detecting differential expression when the number of replications is limited (one or two) [Zhang et al., 2006; Albrecht et al., 2010; Mohammadi et al., 2010; Geiger et al., 2012; Krasny et al., 2016; Casey et al., 2017]. And, the analytical methods used in this study also toke the limitation of the number of replications in consideration. To sum up, we selected two biological replicates of the three HPEV samples to consider both data credibility and experimental cost.

REFERENCES

Albrecht D, Kniemeyer O, Brakhage AA, Guthke R. Missing values in gel-based proteomics. Proteomics 2010, 10(6): 1202–1211.

Casey TM, Khan, JM, Bringans SD, Koudelka T, Takle PS, Downs RA, Lipscombe RJ. Analysis of reproducibility of proteome coverage and quantitation using isobaric mass tags (iTRAQ and TMT). Journal of proteome research 2017, 16(2): 384–392.

Geiger T, Wehner A, Schaab C, Cox J, Mann M. Comparative proteomic analysis of eleven common cell lines reveals ubiquitous but varying expression of most proteins. Molecular & Cellular Proteomics 2012, 11(M111): 014050.

Krasny L, Paul A, Wai P, Howard BA, Natrajan RC, Huang PH. Comparative proteomic assessment of matrisome enrichment methodologies. Biochemical Journal 2016, 473(21): 3979–3995.

Mohammadi M, Anoop V, Gleddie S, Harris LJ. Proteomic profiling of two maize inbreds during early gibberella ear rot infection. Proteomics 2011, 11(18): 3675–3684.

Zhang B, VerBerkmoes NC, Langston MA, Uberbacher E, Hettich RL, Samatova NF. Detecting differential and correlated protein expression in label-free shotgun proteomics. Journal of proteome research 2006, 5(11): 2909–2918.

Results

  1. Results – Lines 181-182 “The mass of the most identified proteins ranged between 10 and 60 kDa” but fig 3a- SDS electrophoresis on 10% separating gel – majority of bands is above 100 kDa – how this discrepancy could be explained?

Author response: Thanks to reviewer for this comment. In this study, SDS-PAGE analysis for partial solution of the extracted HpEVs proteins, and the purpose of this experiment was to judge the quality of HpEVs proteins and the similarity between replicates. Then the proteomic analysis was performed as another experiment, and the proteins were digested for further liquid chromatography with tandem mass spectrometry (LC-MS/MS) characterization in advance. Then the secondary mass spectrometry data was retrieved using Maxquant (v1.5.2.8) according to retrieval parameters, in which the minimum length of peptides was set to 7 amino acid residues and the FDR for protein prediction and identification and PSM identification was set to 1%. Therefore, we hold the point that the identified proteins based on MS/MS base data maybe protein fragments of the protein in the 10% separating gel.

  1. A question – what is the difference in EV proteome compared to the cellular proteome of the respective growth stage? Are EVs simply “buds” of the cells or they are expected to be enriched with some specific cargo proteins? It will be interesting to see how the reported EV proteome is related to the cellular proteome of the respective growth stage.

Author response: Thanks to reviewer for this comment. As we known, EVs carry a rich biologically active cargo of proteins, nucleic acids, lipids and metabolites. It has been established that, the process of cargo sorting of EVs is not considered to be a random event in most cases, which also evolves to a certain extent with the global alterations that transpire within the cells [Anand et al., 2019; Waury et al., 2024]. The protein cargo sorting into EVs follows specific mechanisms which are dependent on ESCRT, tetraspanins and lipids, and some of these proteins are reminiscent of the mechanisms of biogenesis [Boukouris et al., 2015]. Selective mechanism of protein cargo sorting is also known to be controlled by various post-translational modifications (PTMs), which are precise alterations of the proteins after translation enabling them to be either in an active or inactive state [Knorre et al., 2009]. Based on the above results, we also suppose that HpEVs were not simply “buds” of the H. pluvialis cells, and they are expected to be enriched with some specific cargo proteins, for example, the proteins localized in the chloroplast (45.34%), cytoplasm (23.45%), nucleus (10.79%). And, the H. pluvialis sort parent-cell origin proteins into the HpEVs with nature and functions consistent with the microalga cells responding to cell stage or cultivation conditions, thereby endowing these vesicles with cell type-specific nature and biological functions of different growth phase of this microalga. We have added relevant information in the discussion section of the revised manuscript on page 17 and 18.

We completely agree that further work related to the comparison of EV and cellular proteome of the respective growth stage to be conducted. We also plan to extend our studies on HpEVs in more depth according to this suggestion.

REFERENCES

Anand S, Samuel M, Kumar S, Mathivanan S, Ticket to a bubble ride: Cargo sorting into exosomes and extracellular vesicles. Biochimica et Biophysica Acta (BBA) - Proteins and Proteomics 2019, 1867(12) 140203: 1–40.

Boukouris S, Mathivanan S. Exosomes in bodily fluids are a highly stable resource of disease biomarkers. Proteomics Clinical Applications 2015, 9 (3-4): 358–367.

Knorre DG, Kudryashova NV, Godovikova TS. Chemical and functional aspects of posttranslational modification of proteins, Acta Naturae 2009, 1 (3): 29–51.

Waury K, Gogishvili D, Nieuwland R, Chatterjee M, Teunissen CE, Abeln S. Proteome encoded determinants of protein sorting into extracellular vesicles. Journal of Extracellular Biology 2024, 3(1, e120): 1–16.

Comments on the Quality of English Language

The English of the manuscript needs special attention, particularly the verb forms and tenses. For example

  1. Abstract lines 9-11 “However, utilization of these EVs is challenging due to their molecular composition remain poorly understood, resolvent of their composition could provide novel insights into biological properties and application characteristics researches” – understandable but should be edited in proper English (composition which remains, resolution/unraveling instead of resolvent, application characteristics researches – unclear, the last word is unnecessary.

Author response: Thanks to reviewer for this valuable comment. The reviewer is correct, we have checked the manuscript and rewrote the Abstract in the revised manuscript. And the whole text was revised with the help of an English supervisor who graduated from North Dakota State University and worked in USA for 10 years. Please check the revised manuscript.

  1. Line 16 – various instead of variant; lines 17-18 – “Thereby endowing HpEVs with early growth stage contain more proteins associated with cellular functions of primary metabolite” – “endowing” is unnecessary, HpEVs at early growth stage,  cellular functions of primary metabolite – unclear;

Author response: Thanks to reviewer for this valuable comment, which is highly appreciated. We apologize for the poor English of the manuscript, and we have corrected these sentences to make the language more accurate and more appropriate in the Abstract section on line 17 to 20 as “It was revealed that HpEVs from the early growth stage of H. pluvialis contain more proteins associated with cellular functions involving in primary metabolite, cell division and cellular energy metabolism, while HpEVs from the late growth stage of H. pluvialis were enriched in proteins involved in cell wall synthesis and secondary metabolism.” Please check the revised manuscript.

  1. line 20 – “provides the promising role” – unclear, maybe – proves instead of provides

Author response: Thanks to reviewer for this constructive suggestion. We have corrected this sentence as “providing important information in the development and production of functional microalgal-derived EVs.” in line 21 to 22 in the revised manuscript. Please check the revised manuscript.

Lastly, we scrutinized the whole manuscript to avoid language errors. In addition, we consulted a professional editing service and asked several colleagues who are native English speakers to check the English, and made corresponding revisions. Once again, thanks very much for the reviewer’s insightful comments and suggestions.

Round 2

Reviewer 1 Report

Comments and Suggestions for Authors

Accepted

Comments on the Quality of English Language

Minor checks